# Analysis of Tribological Properties of Powdered Tool Steels M390 and M398 in Contact with Al_2_O_3_

**DOI:** 10.3390/ma15217562

**Published:** 2022-10-28

**Authors:** Zbynek Studeny, Michal Krbata, David Dobrocky, Maros Eckert, Robert Ciger, Marcel Kohutiar, Pavol Mikus

**Affiliations:** 1Department of Mechanical Engineering, Faculty of Military Technology, University of Defence, 612 00 Brno, Czech Republic; 2Faculty of Special Technology, Alexander Dubcek University of Trenčín, 911 06 Trenčín, Slovakia

**Keywords:** coefficient of friction, powder tool steel, Al_2_O_3_ ceramic, wear, extruder

## Abstract

The present article examines special steels used for the production of injection screws in the plastic industry, with a glass fiber content of up to 30%. Experimental materials, M390 and M398, are classified as tool steels, which are produced by powder metallurgy-HIP methods (hot isostatic pressing). The main goal of the presented paper is to propose the optimal tempered temperature of M398 steel and also to compare the tribological properties of both materials and to determine the degree of their wear depending on their final heat treatment. Partial results refer to the analysis of hardness, roughness, the overall wear mechanism, the change in the volume of retained austenite due to the tempering temperature, and the EDS analysis of the worn surfaces in individual contact pairs. A ceramic ball Al_2_O_3_ in the α phase was used as the contact material, which had a diameter of 6.35 mm. The ceramic ball performed a rotational movement on the experimental material surface at an elevated temperature of 200 °C using the dry ball-on-disk method. It was experimentally shown that the new M398 material can fully replace the M390 material because it exhibits significantly better tribological properties. The M398 material showed more than a 400% reduction in wear compared to the M390 material. The ideal heat treatment consisted of cryogenic quenching to −78 °C and a tempering temperature of 400 °C. At tempering temperatures of 200 and 400 °C, adhesive wear occurred, which was combined with abrasive wear at a tempered temperature of 600 °C. The averaged coefficient of friction (COF) results show that the M398 material presents less resistance in the friction process and its values are approximately 0.25, while the M390 material showed a COF value of 0.3 after the cryogenic hardening process. The friction surface roughness of the M398 materials also showed lower values compared to the M390 material by approximately 35%. Both of these results are related to the content of M_7_C_3_ and MC carbide particles based on Cr and V in the bulk of the material, which are in favor of the M398 material.

## 1. Introduction

At present, the most progressive method of steel production is the process of powder metallurgy. From an energy and economic point of view, the powder metallurgy process is several times more demanding than the classic methods of steel production. Properties, such as hardness, strength, corrosion resistance, and wear resistance, are several times higher in these steels compared to classically produced steels. The process of steel production by powder metallurgy includes melting basic materials, atomization (production of powders), and subsequent sintering, as detailed in other papers [1,2].

The experimental steels that we will compare in the presented paper are classified as chrome martensitic steels with a high proportion of carbides. Their designations are M398 and M390, and they are manufactured by Böhler. A potential use for this steel is to replace the older M390 steel in the plastic industry, specifically as a material to produce screw conveyors. The screws are exposed to elevated temperatures, mechanical stress, as well as aggressive chemical environments and wear during operation. The implementation of M398 steel in the production of screws is expected to have an effect in the form of increased operational capability as well as production in terms of the use of plastics with a higher content of glass and carbon fibers, which are the main factors influencing the wear and later degradation of screws themselves.

The examined steel has a high chromium content, which results in the formation of carbides, primarily chromium-based M_7_C_3_ type carbides and vanadium-based MC carbides. The occurrence of carbides in the structure of materials generally inhibits the growth of austenitic grain, which has a positive effect on the resulting mechanical properties, as confirmed by the authors’ papers [3,4,5]. The authors refer to the retardation of grain growth by second-phase particles, known as Zener pinning. This phenomenon has a well-established industrial application, such as the suppression of grain coarsening. Since experimental steels contain more than 27.5% and 20.5% of carbides in the case of M390 steel, we can assume that grain growth will not depend on the cooling rate or only to a minimal extent, as confirmed by Cheng et al. and Van Rooyen et al. [6,7]. Farayibi P.K. et al. [8] discussed in another study the possibilities of extending the service life by thickening the X190CrVMo20-4-1 martensitic tool steel powder on two different martensitic stainless-steel substrates by means of a supersolid liquid phase (SLPS) sinter. They assessed the effect of differences in the composition of martensitic stainless steels used as substrates on interfacial diffusion, microstructure, hardness, and bond strength of steel sheets (cladding). Wang T. et al. [9] investigated changes in the microstructure and mechanical properties of martensitic stainless steel M390 at different rolling and annealing temperatures after hot rolling. Their findings led to the conclusion that, with increasing annealing temperature after hot rolling, the plasticity increases and the strength of the material decreases. Additionally, the number of carbides decreases with increasing annealing temperature, but their size increases. Priyadarshini, M. et al. [10] focused on the type of special tool steel AISI P20 used to produce injection molds. They investigated changes in its microstructure and phases using six different heat treatment procedures, where after the austenitization temperature, the given tool steel was cooled to −50 °C, −100 °C, and −150 °C. It was found that this investigated steel cooled to −150 °C had a higher surface hardness compared to processes at the higher mentioned temperatures.

The first main objective of this paper is to propose the optimal tempering temperature of M398 steel in relation to obtaining the lowest material wear rate with and without the use of cryogenic hardening. The second main objective of this paper is to compare the tribological properties of both materials and determine the degree of their wear depending on their final heat treatment. Both of these main objectives are supplemented by additional partial results regarding hardness, roughness, overall wear mechanism, change in the volume of retained austenite due to tempering temperature, and EDS analysis of worn surfaces in individual contact pairs. The article experimentally predicts how tempered temperatures affect the resulting wear of both investigated materials and provides a comprehensive view of wear mechanics. This information can be successfully used in the thermal process of injection of screw production. 

## 2. Materials and Methods

The experimental materials used in this contribution were two special tool steels manufactured by Böhler, namely M390 and M398. Steels are produced using powder metallurgy with the HIP (Hot Isostatic Pressing) method [11]. The description of the production process is described in the publication by Ciger et al. [12]. Due to the production method and chemical composition, the steel provides extremely high resistance to mechanical wear as well as corrosion resistance. The prerequisite for the use of steel is the production of screws for injection molding machines. The main concept for increasing the macro-hardness is the high content of MC and M_7_C_3_ carbides, which can be observed in the microstructure itself in Figure 1. The basic microstructure of test tool steels was investigated by scanning electron microscopy (SEM, Tescan Vega 3, Tescan Orsay Holding, Brno, Czech Republic). The samples in this case were heat-treated (Q + T) to the highest hardness. 

These steels have their uses in the plastics industry, specifically to produce injection molding screws. The helix performs a rotational movement which moves the solid granulates from the hopper into the space in front of the screw. It is gradually heated and melted to the injection temperature depending on the type of plastic polymer in the temperature range of 160–250 °C. The injection of plastic takes place by means of a straight movement of the screw to the front position. The back flow cylinder prevents the backflow of the molten plastic granulate. The investigated material M398 is a newly developed material, the task of which will be to replace material M390. Other properties that predetermine M398 steel to produce screws include, high dimensional stability during heat treatment, good resistance to chemical corrosion, and the possibility of polishing to a mirror finish. 

Table 1 shows and compares the values of the prescribed chemical composition from Böhler and the chemical composition measured by the authors using a Spectrolab Jr. chemical analyzer. The basic mechanical properties are listed and compared in Table 2. The manufacturer did not provide the exact chemical range of the individual elements.

In the processes of all realized experiments, dry sliding friction on powdered tool steels M390 vs. M398 and comparison of wear in contact with ceramic bearing ball Al_2_O_3_ at a constant measuring temperature of 200 °C were investigated. The whole measurement process took place on an instrument, the UMT TriboLab (Bruker Austria GmbH, Wien, Austria), where the main changing parameter was the different tempered temperatures of the samples (200 °C, 400 °C, 600 °C). Half of each material was also cryogenically turbid to reduce the residual austenite. The marking of the samples consisted of the marking of the experimental material and the subsequent tempering temperature. The samples are marked with the type of steel, and the subsequent number determines the tempering temperature of the samples. The letters DF (deep freezing) indicate samples that have undergone a cryogenic hardening process. Samples that did not have DF at the end were quenched to 50 °C and then tempered to the prescribed temperatures written on their labels. Wear results and roughness measurements were measured by light optical microscopy (LOM) and atomic force microscopy (AFM, Oxford Instruments, Abingdon, UK), respectively. Using the AFM microscope, 3D topographies of the ceramic balls, as well as the formed grooves, were also obtained, and their final texture surface was determined after individual experiments. All the obtained results for the coefficient of friction and wear are discussed in the next part of the article.

### 2.1. Heat Treatment

The experimental materials were supplied in the form of a bar with a diameter of 50 mm and a length of 500 mm. The materials were delivered in a soft annealed condition with a maximum hardness of 280 HB for M390 steel and 330HB for M398 steel. For experimental work, samples from both materials were heat-treated according to Figure 2. The samples were heated to 1150 °C and then quenched in oil. Six samples were prepared for each steel, of which three samples were hardened and subsequently tempered to a temperature of 200, 400, and 600 °C (Figure 2a). The remaining three samples were quenched, followed by cryogenic cooling to −78 °C with dry ice, followed by tempering, such as the previous samples. Two thermocouples (platinum and tungsten) were welded to the samples. Depending on the temperature, small amounts of electricity is generated at the ends of the conductor’s voltage for accurate temperature monitoring during cryogenic hardening. The graphical course of curing with freezing is on (Figure 2b).

### 2.2. Evaluation of Hardness and Roughness

Hardness was measured using the Vickers method on an Instron Wolpert 930 (Norwood, MA, USA) under a load of 49.05 N [14]. The results were averaged over five values for each sample. Hardness verification was measured on the experimental material and on the Al_2_O_3_ pressure ball. For its qualitative determination, the hardness measurement was always performed at five places, from which the average value was determined [15]. The surface texture of the pressure balls and tested tool steels was evaluated by AFM microscopy. The measurements were carried out at five points each time the tip moved on the surface of the pressure balls and the experimental samples. As the main roughness parameter was used, the arithmetical mean height (Sa) [16]. 

### 2.3. Dry Sliding Test

Tribological tests were performed using a universal tribological device, the UMT TriboLab from Bruker (Billerica, MA, USA). The measurement method was ball-on-disk. All measurements were performed in a thermal chamber. The chamber is equipped with thermocouples, which are used to precisely control the set temperature in the process of tribological measurement. One type of contact material was used in the ball-on-disk measurement method. This material was Al_2_O_3_ with a diameter of 6.35 mm.

Rotational tribometric tests were performed at an above-average load of 40 N. The extruded ceramic ball Al_2_O_3_ was firmly clamped in the holder and slid onto the tool steel at a pressed speed of 0.34 m/s over a diameter of 13 mm without using lubrication. All measurements were performed at an elevated temperature of 200 °C and the measurement time was 1800 s. Samples of tool steels were created in the shape of a disk with a diameter of 50 mm and a thickness of 9 mm. 

The volume loss of the test material was measured using a TalySurf CLI 1000 (Berwyn, PA, USA). The average volume loss value was obtained from three locations on the worn track (at angles of 120°, 240°, and 360°). The UMT Tribolab tribometric instrument (Bruker, Billerica, MA, USA) automatically calculates and records the coefficient of friction (COF) with the help of electrical sensors using the ratio of the pressing axial force (Fx) and the horizontal radial force (Fz) [13,17].

## 3. Results and Discussion

### 3.1. Effect of Heat Treatment on the Amount of Residual Austenite and Tempering Diagrams

Using the saturation magnetization method (SMM), we obtained the proportion of volume fractions of retained austenite. By cooling to negative temperatures, a reduction in the amount of retained austenite was achieved. However, this reduction could also be achieved by tempering, as can be seen in Figure 3. Here we observe that at a tempered temperature of 540 °C there is a significant decrease in the proportion of retained austenite to less than 1%. 

Experimental powder steels can be heat treated for increased corrosion or abrasion resistance. For this reason, it is necessary to choose the correct temperature range during heat treatment for the final part. Tempered temperatures in the range of T = (200 ÷ 300) °C are suitable for both steels when the material is designed for high corrosion resistance. Wear resistance for material that has not been quenched to subzero temperatures is in the temperature range of 540–560 °C (Figure 4a). For materials that have been frozen, this area shifts between temperatures of 510 and 530 °C, as can be seen in Figure 4b.

### 3.2. Comparison Hardness of Powder Tool Steels M390 and M398 after Tempering

The samples prepared and heat treated according to Section 2.1 were used to measure the hardness using a Vickers hardness tester and a HV10 load. The measured hardness values are given in Table 3. From the measured hardness values, we can observe a higher or comparable hardness of M398 steel in comparison with M390 steel. The use of freezing and hardening has been included to reduce the amount of retained austenite in M390 steel. Freezing and hardening according to the measured hardness values do not have a significant effect on the hardness of M398 steel, as evidenced by the statement of the steel manufacturer [18]. The amount of retained austenite in M398 steel decreases with increasing tempered temperature. We observed the highest hardness of steel M390 and M398 after tempering at a temperature of 400 °C using freezing and hardening. The increase in each hardness compared to a tempered temperature of 200 °C is due to the formation of a secondary hardness during tempering. The effect of freezing and hardening in the case of M390 steel is demonstrable by increasing the hardness during tempering to a temperature of 400 °C, where the hardness increased from 717 (HV10) to 833 (HV10), which represents an increase of 16.1%. In the case of M398 steel, there was no significant increase in hardness as the steel showed high hardness even after hardening without freezing. This fact proves that M398 steel is a more suitable choice for applications where cryogenic hardening is not possible. The material retains high hardness values up to a tempered temperature of ~500 °C. At a tempered temperature of 600 °C, there is a significant decrease in hardness due to the hardening process. The lowest hardness of 455 (HV10) was measured on M390 steel after the hardening process without freezing. 

### 3.3. Roughness of Al_2_O_3_ Balls and Tool Steel

The average roughness result of the pressure balls is shown in Figure 5b. The surface texture value Sa for the ceramic ball Al_2_O_3_ was Sa = 0.03 µm. The pressure balls showed very low roughness values, which is associated with their production process and the possibilities of use in bearings. 

The surface texture of the experimental tool steels M390 and M398 together with their average values of the Sa parameter are shown in Figure 5a. The samples were created on a turning machine, where finishing turning was used during the last machining cycle. Subsequently, all samples were ground to a single fixture using a magnetic grinder. The resulting roughness of tool steels M390 and M398 had the same values, and their measured value was Sa = 0.68 µm.

### 3.4. Wear Comparison of Materials M390 and M398

Volume loss in mm^3^ was measured on samples of experimental materials M390 and M398 using a TalySurf CLI 1000 profilometer. The resulting volume loss did not consider the proportion of extruded material (Figure 6) at the edges of the friction surface. The extruded material at the edges of the friction surfaces occurred only in some measurements and was processed separately in the graphs. The volume loss of the Al_2_O_3_ test ceramic ball material was also calculated. The calculation of the volume loss of the worn ball material is:(1)$$V\;=\;\pi \;\cdot \;h^{2}\;\cdot \;\left(r-\frac{h}{3}\right)$$
where *h* is the height of the worn ball part and *r* is the radius of the ball. The Al_2_O_3_ ceramic material has a very high hardness value of 1447 HV10 compared to the hardness of the M390 and M398 experimental tool steels, which range from 455 (HV10) to 806.5 (HV10) depending on the heat treatment. This difference in the hardness of the contact materials represents a significant increase in hardness in favor of the Al_2_O_3_ ceramic ball. Thus, it can be assumed that the main part of the friction mechanism in the form of wear will occur on experimental steels M398 and M390.

A comparison of volume losses of selected powdered tool steels is shown in Figure 7. It can be observed, and from the obtained results, that the change in heat treatment significantly affects the wear of experimental materials. The given figures show the worn areas from the top view and their cross-sections. The red color in the cross-sections represents the removed part of the material after the friction process. The green color shows the plastically deformed material, which is extruded on the edges of the friction surfaces. In the first stage, we will compare the wear of experimental material M390 after the final heat treatment (cryogenic hardening and tempering) as well as samples that were only hardened and tempered without the process of cryogenic hardening. The highest volume loss of all experimental samples of M390 material was observed at a tempered temperature of 400 °C and a cryogenic hardening process (Figure 7b). The wear reached 0.442 mm^3^, very similar to the wear result that occurred at a tempered temperature of 200 °C, and its value was 0.441 mm^3^ (Figure 7a). The lowest wear from a given series of samples of experimental material M390 was reached at a tempered temperature of 600 °C and reached a value of 0.301 mm^3^ (Figure 7c). Samples of M390 material that underwent a heat treatment process without cryogenic hardening showed slightly lower values of volume losses. From a given series of samples, the highest rate of wear was again measured in the sample that was tempered at 400 °C and reached a value of 0.396 mm^3^ (Figure 7e). The sample reached a wear of 0.346 mm^3^ at a tempered temperature of 200 °C (Figure 7d). The lowest wear from a given series of all samples of M390 material was reached at the highest tempered temperature of 600 °C and reached a value of 0.301 mm^3^ (Figure 7f). The second stage compares the wear of experimental material M398 that underwent the same heat treatment as material M390. The given experimental material after heat treatment with cryogenic hardening showed the highest wear rate at the highest tempered temperature of 600 °C (Figure 7i). The wear of the given sample reached a value of 0.125 mm^3^. It is clear from the results that at tempered temperatures of 200 and 400 °C, very similar results in volume losses of the material after the friction process were achieved. Figure 7g shows the friction surface at 200 °C with the wear value reaching 0.101 mm^3^. The best results for volume losses in each series of samples were achieved at a temperature of 400 °C, where the wear was approximately 0.096 mm^3^ (Figure 7h). In the last series of M398 samples that underwent a hardening and tempering process without cryogenic treatment. The wear result was different due to the tempered temperatures. For the given samples, the highest rate of volume loss was observed on the sample that was tempered at the lowest temperature, 200 °C (Figure 7j). The wear reached 0.181 mm^3^. The tempered sample at 600 °C showed better wear results and had a volume loss value of 0.142 mm^3^ (Figure 7l). The lowest value of volume losses was achieved by the sample, which was tempered at 400 °C and the wear value was 0.096 mm^3^ (Figure 7k). All samples that were tempered at 600 °C also showed a higher degree of plastic deformation in the form of moving the extruded material from the friction groove to its edge. This plastic deformation also occurred in other samples, but to a much lesser extent. Its evaluation will be presented in the next part of the paper. An overall comparison of the volume losses of both experimental materials M390 and M398 is shown in Figure 8. In the given comparison, the plastic deformation of the worn material in the form of extruded material along the edges of the samples was not considered. It is clear from the results that the M390 material showed significantly worse wear results compared to the M398 material. 

Figure 8a shows the results of a comparison of volume losses of materials that have undergone a final heat treatment using cryogenic hardening. At temperatures of 200 and 400 °C, material M390 achieved more than a 400% increase in wear compared to material M398. At a maximum tempered temperature of 600 °C, this increase was approximately 300%. Hardness results from Table 3 show very similar values after the individual tempered temperatures. The matrix of both materials was formed by a martensitic structure, with the difference that the total content of carbide particles is different. M398 contains approximately 30% carbides, in contrast to M390, where there is approximately 20%. Another reason for the lower wear of the M398 material is the proportion of individual types of carbides. M398 contains approximately 5% of MC carbides compared to M390, which has approximately 2.5%. These carbides are V-based and have a hardness of approximately 2800 HV [19,20]. The reason for this higher proportion of MC carbides is because of the higher proportion of V in the chemical composition of the M398 material. M_7_C_3_ carbides are Cr-based, and their hardness is lower compared to MC carbides, and they have values in the range of 1500–2000 HV [21,22]. Comparing the results of experimental materials, M390 and M398, that did not undergo the cryogenic cooling process in Figure 8b, we found similar results as in the previous case. This means that M398 steel showed better wear results than M390 steel. At a tempering temperature of 400 °C, material M390 again showed more than a 400% increase in volume losses compared to material M398. At a tempered temperature of 200 °C, the M390 material had an almost 200% higher wear rate. The last samples that were tempered at 600 °C had approximately the same wear rate, with M390 having only a 14% higher wear rate than M398. Thus, the results clearly indicate that the experimental steel M398 shows much better wear results compared to the steel M390. The ratio of retained austenite to martensitic matrix varies depending on the tempering temperature [23]. At the tempered temperature, the proportion of retained austenite in both materials, M390 and M398, is only approximately 1%; at a tempered temperature of 200 °C, it is approximately 25% in M390 and in M398, its content is approximately half as much as the 12% retained austenite. This retained austenite is depleted of carbon at the tempered temperature and is converted to ferrite and cementite in the case of steels with increased carbon content [7,24,25]. The process of cryogenic hardening in the heat treatment of M398 steel leads to its transformation into ferrite, and thus, to the formation of the so-called two-component microstructure, which consists mainly of martensitic matrix and ferrite (+carbides). Subsequent tempering at higher temperatures leads to its transformation, but at the same time, it increases the toughness of the martensitic matrix. Therefore, it is ideal for use in the friction process of a given steel at a temperature of 400 °C. At a temperature of 400 °C, we also achieved a comparable wear rate for the sample that was not cryogenically hardened. The reason for this result is probably the ratio of residual austenite and tempered martensitic matrix.

### 3.5. Comparison of Al_2_O_3_ Pressure Ball Wear

The comparison of volume losses was also performed on Al_2_O_3_ balls, which were created from Al_2_O_3_ material. The overall comparison is shown in Figure 9. From the results, it can be observed that when comparing the contact pairs, the ceramic ball was more worn in all cases in contact with material M390. Figure 9a shows the results of a comparison of volume losses of friction balls with experimental materials that have undergone a final heat treatment using cryogenic hardening. The highest wear of the ceramic ball in the case of contact with the M390 material occurred at a tempered temperature of 400 °C. The wear value was 0.112 mm^3^. Subsequently, a slightly lower wear occurred at a tempered temperature of 200 °C, and in this case, the wear was 0.104 mm^3^. Significantly, the lowest wear occurred at the highest tempered temperature of 600 °C, where the wear value was at least 0.004 mm^3^. For M398 materials, the Al_2_O_3_ wear values decreased significantly. The wear of the ceramic balls on contact with the M398 material ranged from 0.006 mm^3^ in the case of contact with the material tempered at 400 °C up to 0.002 mm^3^ in the case of a sample which was tempered at 600 °C. At temperatures of 200 and 400 °C, there was an average 18-times increase in wear compared to M398. Higher wear values were associated with the wear of experimental materials. Figure 9b shows the results of comparing the volume losses of friction balls with experimental materials that had undergone a final heat treatment without the use of cryogenic hardening. Here, the overall results are similar to those in Figure 9a. Additionally, the highest wear of the Al_2_O_3_ ceramic ball was achieved in contact with the M390 material at a tempered temperature of 400 °C, and its value was 0.104 mm^3^. This was followed by a decrease in wear at 200 °C, resulting in a wear of 0.008 mm^3^. The lowest wear value, which is comparable to both experimental materials, occurred at a tempering temperature of 600 °C. At 200 °C, this was an almost 8-times increase in wear compared to M398, and at 400 °C, this difference increased almost 11-times. A significant decrease in wear at the highest tempered temperatures of 600 °C is associated with plastic deformation when the ceramic ball pushes the material to the edges of the friction surfaces due to a significant reduction in hardness, thus, increasing the toughness of samples at a given temperature [26,27]. As already mentioned, the wear of the ceramic balls is linked to the wear results of the individual experimental materials, M390 and M398. The increase in wear also led to an increase in the wear values of the ceramic material. Of course, the type of wear is also affected by its type. This issue will be examined in the wear mechanisms section. Overall, however, the ceramic ball showed significantly lower wear values when in contact with the experimental material M398. Therefore, we can state that the material is more suitable for use in processes where friction processes occur.

### 3.6. Comparison of Extruded Material

The effect of plastic deformation in the form of extruded material on the edges of the friction grooves was also investigated, and the results of the study are shown in Figure 10. The displacement of this material was affected by the resulting hardness of the material. As can be seen, Figure 10a shows the results of a plastically transferred material on the edges of the friction grooves in samples of materials M398 and M390 that have undergone a cryogenic hardening process. The highest degree of plastic deformation occurred in both materials after the highest tempering temperature of 600 °C. At a given temperature, material M398 had an extruded material value of 0.067 mm^3^, while material M390 had a value of 0.039 mm^3^. At tempering temperatures of 200 °C and 400 °C, the plastic deformation process was much smaller. It ranged from 0.01 to 0.016 mm^3^. Therefore, it can be stated that it was almost similar. Referring to Figure 10b, which represents the processes without cryogenic hardening, the values of the plastically extruded material increased linearly, especially for the M390 materials, due to the increasing tempering temperature. Material M398 had values of extruded material at temperatures of 200 °C and 400 °C at only 0.001 mm^3^. At a temperature of 600 °C, the plastically deformed material increased to the edges of the friction surface and reached a value of 0.019 mm^3^. 

The decrease in the hardness of the samples, which is associated with the resulting microstructure, and thus, leads to the formation of softer structural samples, leads to an increase in the plastically deformed material, which moves to the edges of the friction surfaces. As a result, wear occurs, but it has a different character than in the case of harder samples, in which the abraded material is removed from the contact friction surface in the form of adhesive or abrasive wear. This claim will be further examined in the wear mechanisms section.

### 3.7. Coefficient of Friction

A comparison of the coefficients of friction values (COF) of the experimental materials, M390 and M398, is shown in Figure 11. The individual comparison pairs of both materials are compared with respect to the final tempered temperature. The results clearly indicate that the experimental material M398 shows lower values of the coefficient of friction compared to material M390. The average COF value of the M390 material from all tempered temperatures that underwent the cryogenic hardening process was 0.35. For comparison, M398 had an average COF of 0.28 under the same heat treatment conditions. Therefore, the decrease in COF in favor of M398 was 20%. The decrease in COF is again associated with the percentage of carbides as well as their chemical composition in the matrix material [28]. In Figure 11a,b,d,e, we can observe that the COF values of the experimental material M390 stabilized after the initial increase, while material M398 behaved in the opposite way, and after the initial stabilization, there was an increase in COF. This increase is characterized by a change in sliding conditions in the contact zone. The higher proportion of carbide particles and their gradual release is likely to lead to a change in the type of wear, and thus, to a slight increase [29]. At the highest tempered temperature of 600 °C, it was confirmed for both types of materials that material M398 again has a lower COF value compared to material M390. In the case of cryogenically quenched samples (Figure 11c), the difference was only minimal, and the average COF value was 0.25 ± 0.03 for M398, while M390 showed a COF value of 0.3 ± 0.02 (Figure 11f), compared to the previous case. In the given figure, COF jumps in M398 materials, which was caused by plastic deformation of the material. This plastically deformed material was pressed by the ceramic ball in front of it, and thus, increased the COF value abruptly. After the release or transfer of this plastically deformed material, the COF value stabilized.

### 3.8. Wear Mechanisms

Through a detailed examination of the worn surface using LOM and SEM, we identified two types of wear in selected observations. These two types of wear occurred either separately or simultaneously, depending on the method of final heat treatment of the experimental samples [30]. The first type of wear that occurred on worn surfaces was abrasive wear, which created deep scratches in the materials due to peeled hard microparticles [31]. These hard microparticles of the released material move freely and also form cutting wedges, which later form deep grooves directly in the test material or fill the formed microcraters. These released microparticles become harder than the base material due to intense plastic deformation or oxidation by atmospheric oxygen [32,33,34]. The two surfaces that rub against each other are never ideally smooth, and contact does not occur over the entire surface but on a large number of contact points [35]. By the action of forces, the peaks of the protrusions on the surface are plastically deformed and the atoms of both surfaces are in close contact and form the so-called micro-joints. All these micro-joints later break in places above the material contact of one of the friction bodies due to the formation of the surface. This area is reinforced by plastic deformation, and its strength is higher than the strength of subsurface areas. Following the breakage of the microscopes, the microparticles transferred from the surface of one body to another. These subsequently remained attached to the surface of the second body. Alternatively, as loose particles, they move between materials and encourage the formation of additional abrasive particles [36]. The given abrasive wear was observed on both types of material samples, which were tempered to the highest temperature, 600 °C (Figure 12b,d,f,h). As the tempered temperature decreased to 400 °C and 200 °C, the wear conditions subsequently changed to adhesives (Figure 12a,c,e,g). The experimental material was gradually cured to form a hard layer which withstood volumetric wear. Although this hard shell had a higher hardness than the base material, it also increased its fragility. As a result of cyclic loading, microcracks form on the surface of the shell. These microcracks are mostly perpendicular to the axis of the friction mechanism, and when they cross, the worn part, which has the shape of a thin shell, is torn off. These thin shells of material can be gradually crushed and turned into hard abrasive microparticles [37,38].

EDS analysis of worn surfaces of experimental materials, M390 and M398, was also performed on selected samples. The contents of elements O and Al were evaluated from the EDS analysis (Figure 12a,b). In comparison, we observed that for material M390, which underwent a process of cryogenic hardening (Figure 12a), at a tempering temperature of 200 °C, the content of individual elements was in the ratio of 27.68 and 3.74% Al. In contrast, the second sample (Figure 12b), which was tempered at 600 °C, had an element content of 29.21 and 2.15% Al. When comparing the O content, we can notice its different occurrences. On the samples that were tempered at 200 °C, the O resolution is finer and more uniform. However, in the second sample, tempered at 600 °C, the distribution of O was uneven. The occurrence of this type of wear is influenced by several factors [39,40]. Through their mutual combination in the friction mechanism, there can be more types of wear (adhesive and abrasive) between the contact pairs than there was in our case.

For a detailed examination of the friction mechanism, two balls were also selected in which the wear mechanism was observed, which are shown in Figure 13. The first figure shows the area of the Al_2_O_3_ ceramic ball (Figure 13a) which was in contact with the experimental material M390, which was not cryogenically quenched and tempered at 200 °C. In this case, the surface of the friction surface of the ceramic ball is formed by adhesive wear, which is manifested by the occurrence of microcracks, delamination of the surface, as well as signs of uneven surface wear. In addition, when analyzing the second surface of the Al_2_O_3_ ceramic ball (Figure 13b), which was rubbed with M398 material, which had undergone a cryogenic hardening process and was tempered at 600 °C, a different method of wear occurred. In this case, the surface is different and there is a typical occurrence of parallel abrasive grooves and deep craters, which were formed by pushing the microabrasive particles into the material.

### 3.9. Analysis of Surface Roughness

All values of the roughness of friction grooves depending on the type of experimental material are shown in Table 4. The value of the parameter Sa of the basic materials before the friction process after their production was Sa = 0.68 µm. When comparing the roughness values of both experimental materials with respect to the use of the cryogenic hardening process, we can state that the experimental material M398 showed lower values of the parameter Sa of friction grooves compared to material M390. The lowest value of the friction groove Sa was achieved on material M398, which underwent a cryogenic hardening process and was tempered to a temperature of 400 °C. Its roughness value was 0.06 µm. By comparing samples of both experimental materials using the cryogenic hardening process, their average value was reduced by 40% in favor of M398. Subsequently, for samples that did not undergo cryogenic hardening, the results differed slightly. The lowest value of the friction groove parameter Sa was again reached on M398 materials, but this time at the highest tempering temperature, namely 600 °C. The Sa value was only 0.02 μm. This significant decrease in Sa was associated with plastic deformation in each sample, which reached the highest value of all measured samples. The Al_2_O_3_ ceramic ball created a smooth friction track in which the excess material was extruded to the edge of the friction groove due to a significant reduction in the hardness of the sample. By comparing samples of the two experimental materials without using the cryogenic hardening process, their average value was reduced by 65% in favor of M398. Figure 14 shows and compares selected 3D topographic areas. In the given figures, we can optically observe that the experimental material M398 does not show any significant peaks on its surface, and the total area forms a continuous plane. In contrast, M390 materials have significant protrusions as well as deep cracks, which increase the resulting values of the parameter Sa.

In addition to examining the surface texture of the experimental materials, the values of the contact surfaces of Al_2_O_3_ ceramic balls were also measured. The Sa of the ceramic ball before the friction process was Sa = 0.03 µm. Within this analysis, qualitatively worse results were obtained here than on experimental tool steels. A comparison of all measured values of the parameter Sa of the surfaces of the ceramic ball Al_2_O_3_ is also shown in Table 4. The measured values of the values correspond in parallel with most of the results of the values of the experimental materials, M390 and M398. When comparing the values of the parameter Sa of ceramic balls with respect to contact with experimental materials with different heat treatment processes, we can state that the beads that were in contact with material M398 showed lower values of Sa than the balls that were rubbed with material M390. Unlike experimental tool steels, the surface texture of ceramic balls contains many deep craters, as you can see in Figure 15, in which selected 3D surfaces of the surface texture of the Al_2_O_3_ ceramic balls are compared. The intensity of the alternation of peaks and troughs increased with respect to the individual heat treatment conditions under which the experimental steel samples were processed. We observe that the highest values of Sa occurred mainly on samples that were tempered at a temperature of 200 °C. The lowest tempered temperature led to the highest content of internal stresses in the experimental materials, M390 and M398, and thus, a surface release of microparticles from the surface of the materials could occur. For M398 materials, most of the values matched and there were no large fluctuations in the measured values of this parameter on the contact surface of the ceramic ball. Experimental material M390 showed more significant differences in the values of the parameters Sa of the contact ceramic material. In the case where samples of both materials underwent a process of cryogenic hardening and subsequent tempering at selected temperatures, the average value Sa of ceramic balls from all measurements agreed. A different average value occurred for samples without cryogenic hardening, where the average value of the Sa parameter of the ceramic balls reached lower values by approximately 35% in favor of the beads that were in contact with the M398 material.

## 4. Conclusions

In this present study, the tribological properties of two special tool steels, M390 and M398, were produced by the powder metallurgy process. The experimental materials were in contact with ceramic materials based on Al_2_O_3_. Both materials were equally heat treated and tempered to three different temperatures: 200, 400, and 600 °C. In addition to tempering temperatures, a different parameter was the use of cryogenic hardening in half of the samples at a temperature of −78.5 °C. The following conclusions can be drawn from the present work:The hardness of the contact pairs significantly influences the wear process of the pre-sented experimental steels, M390 and M398, because their chemical composition affects the formation of carbide particles that increase hardness and, at the same time, reduce the resulting wear in favor of the M398 material.The highest hardness of material M398 was achieved at a tempering temperature of 400 °C in both types of hardening materials. M390 reached the highest hardness during hardening without freezing at a temperature of 200 °C. Cryogenic hardening has led to an increase in hardness for both steels at approximately 830 HV and is, therefore, recommended in the thermal process for both materials at a subsequent tempering temperature of 400 °C.The M398 material showed more than a 400% reduction in wear compared to the M390 material after cryogenic hardening. The lowest wear values were recorded at tempering temperatures of 200 °C and 400 °C after cryogenic quenching and had a value of approx. 0.1 mm^3^.The 3D surface texture parameter of the worn grooves on the M398 materials showed lower values than for the M390 material. The lowest values of the parameter Sa were achieved on samples that did not undergo the process of cryogenic hardening on material M398 and their average value was Sa = 0.05 µm.The effect of plastic deformation in the form of the extruded material on the edges of the friction surfaces was observed to the greatest extent at the highest tempering temperature of 600 °C. At a given temperature, there was a significant decrease in the hardness of the base matrix in all samples of both types of experimental materials. Therefore, this temperature is no longer recommended for tempering.The results of the coefficient of friction show that material M398 puts less resistance in the friction process and its values are approximately 0.25, while material M390 showed a COF value of 0.3, after the cryogenic hardening process. The difference in COF values was higher for samples that did not undergo a cryogenic treatment process.Increasing the temperature also changes the type of wear. At 200 °C and 400 °C, there was predominantly only purely adhesive wear with small evenly spaced oxidation areas. At a temperature of 600 °C, two types of wear occur: adhesive and abrasive. The oxidizing layers form large continuous areas. These results were the same for both materials.Experimental material M398 can fully replace material M390 in processes where its degradation occurs due to the friction mechanism. The M398 material showed significantly better results in all the values we measured. The inclusion of cryogenic hardening in the production process of M398 material is recommended by us. The ideal tempering temperature for achieving the highest wear resistance is 400 °C. A component composed of M398 can be expected to last several times longer in the working environment, up to 200 °C, than M390. Despite the higher purchase price of the M398 material, it is possible to assume its return and overall savings in operation in the work process cycle, e.g., injection molding machines in the plastics industry.

## Figures and Tables

**Figure 1 materials-15-07562-f001:**
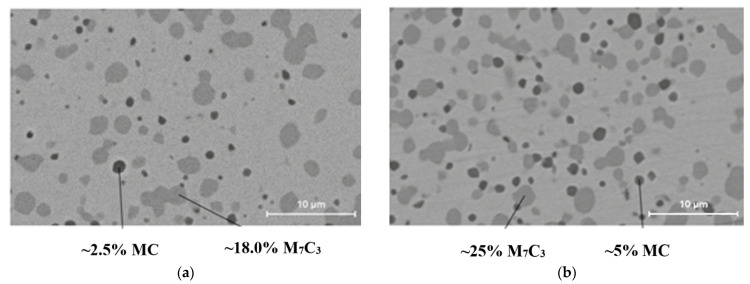
Microstructure of tool steels (Q + T): (**a**) M390; (**b**) M398 [13].

**Figure 2 materials-15-07562-f002:**
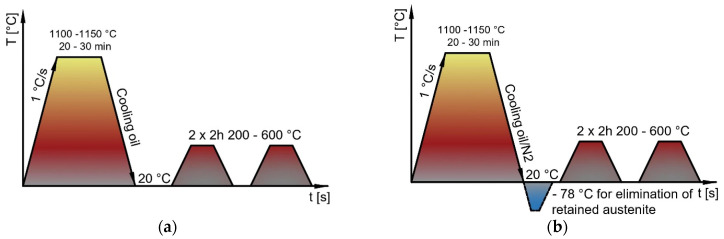
Schematic diagram of thermal processes of M390 and M398 steels: (**a**) without deep freezing; (**b**) with deep freezing.

**Figure 3 materials-15-07562-f003:**
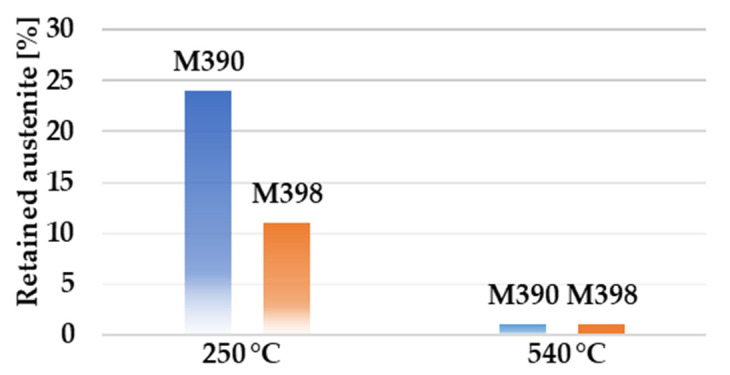
Dependence on tempered temperature on the amount of residual austenite.

**Figure 4 materials-15-07562-f004:**
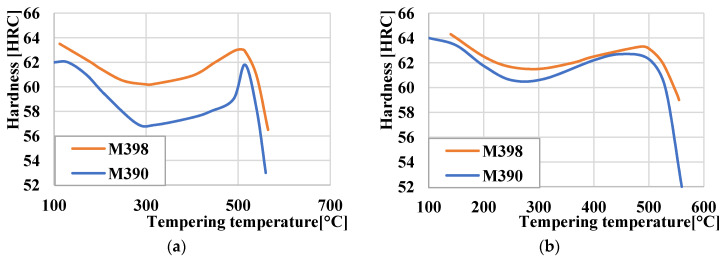
Tempering diagrams: (**a**) without deep freezing; (**b**) with deep freezing.

**Figure 5 materials-15-07562-f005:**
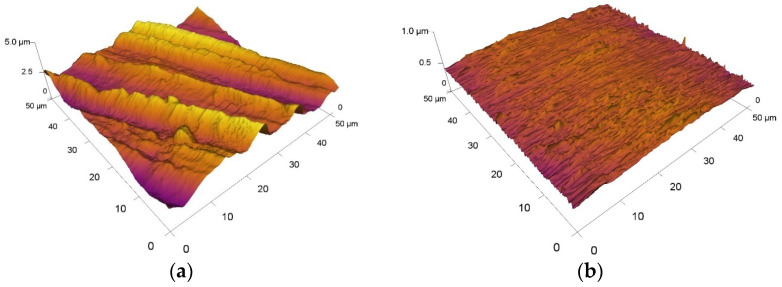
Surface texture evaluated by the AFM: (**a**) M390-M398—Sa 0.68 µm, (**b**) Al_2_O_3_-Sa 0.03 µm.

**Figure 6 materials-15-07562-f006:**
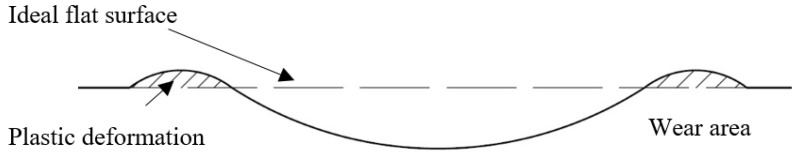
Marked area of removed and printed material.

**Figure 7 materials-15-07562-f007:**
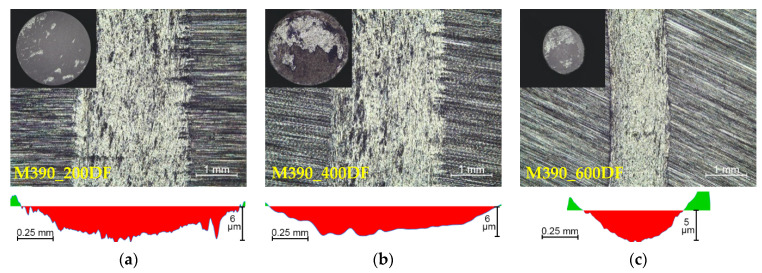
Worn surface of the sample: (**a**) M390_200DF, (**b**) M390_400DF, (**c**) M390_600DF, (**d**) M390_200, (**e**) M390_400, (**f**) M390_600, (**g**) M398_200DF, (**h**) M398_400DF, (**i**) M398_600DF, (**j**) M398_200, (**k**) M398_400, and (**l**) M398_600.

**Figure 8 materials-15-07562-f008:**
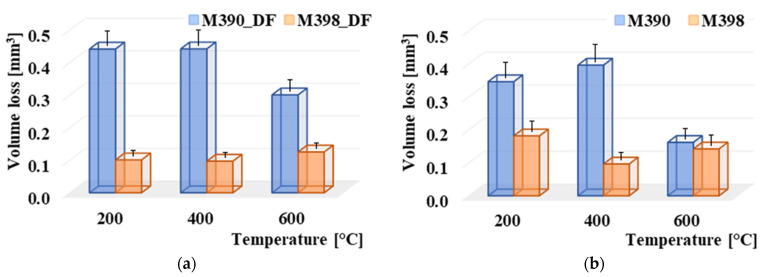
Comparison of wear of materials M390 and M398: (**a**) wear of materials with cryogenic treatment; (**b**) wear of materials without cryogenic treatment.

**Figure 9 materials-15-07562-f009:**
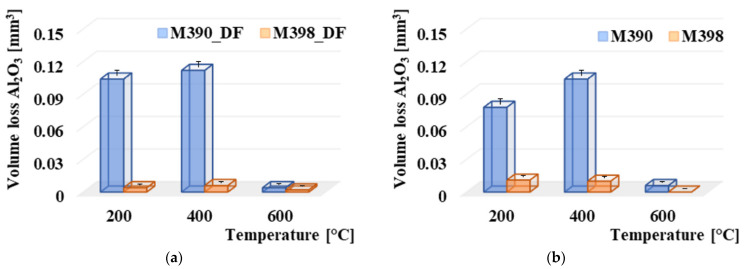
Comparison of volume loss Al_2_O_3_: (**a**) volume loss Al_2_O_3_ with M390_DF and M398_DF and (**b**) volume loss Al_2_O_3_ with M390 and M398.

**Figure 10 materials-15-07562-f010:**
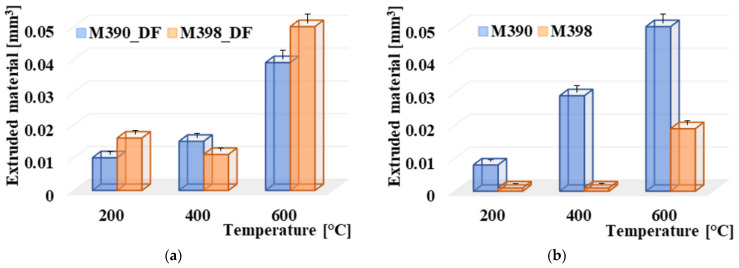
Comparison of plastic deformation results: (**a**) M390_DF and M398_DF and (**b**) M390 and M398.

**Figure 11 materials-15-07562-f011:**
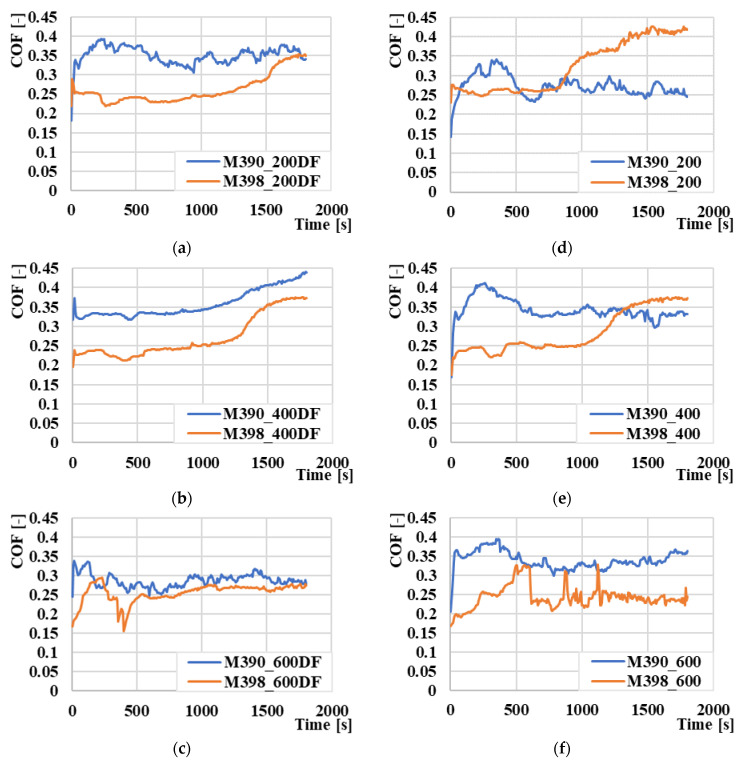
Comparison of the results of the friction coefficient values for: (**a**) M390_200DF and M398_200DF, (**b**) M390_400DF and M398_400DF, (**c**) M390_600DF and M398_600DF, (**d**) M390_200 and M398_200, (**e**) M390_400 and M398_400, and (**f**) M390_600 and M398_600.

**Figure 12 materials-15-07562-f012:**
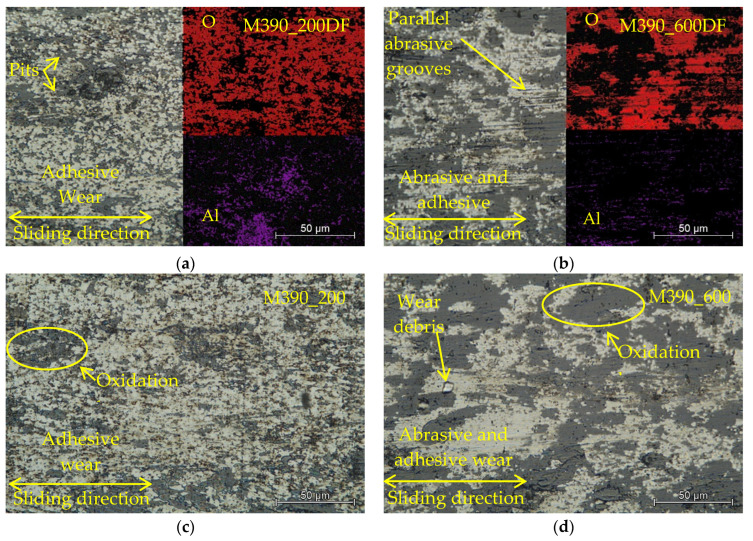
Wear analysis of worn surfaces M390 and M398: (**a**) M390_200DF, (**b**) M390_600DF, (**c**) M390_200, (**d**) M390_600, (**e**) M398_200DF, (**f**) M398_600DF, (**g**) M398_200, and (**h**) M398_600.

**Figure 13 materials-15-07562-f013:**
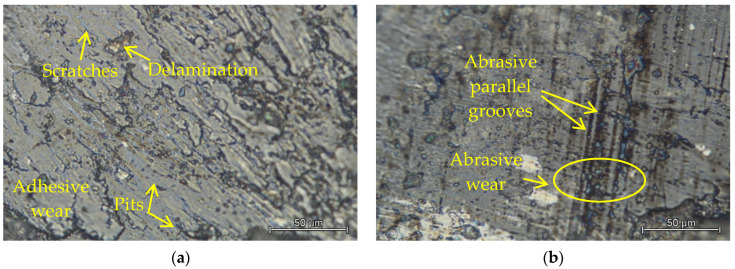
Wear analysis of worn surfaces Al_2_O_3_ with contact: (**a**) M390_200 and (**b**) M398_600DF.

**Figure 14 materials-15-07562-f014:**
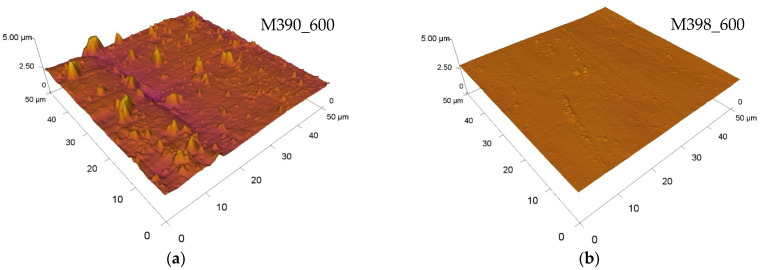
Three-dimensional comparison of material roughness values of M390 and M398: (**a**) M390_600 and (**b**) M398_600.

**Figure 15 materials-15-07562-f015:**
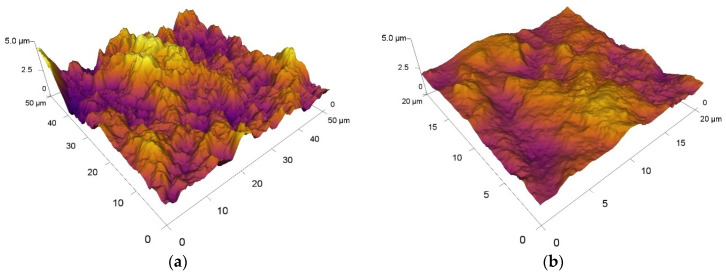
Three-dimensional comparison of the roughness values of the ceramic ball Al_2_O_3_ in contact with: (**a**) M390_600 and (**b**) M398_600.

**Table 1 materials-15-07562-t001:** Chemical composition of the investigated M390 and M398 steels (wt. %).

Element	C	Si	Mn	Cr	Mo	V	W
BOHLER M390	1.90	0.70	0.30	20.00	1.00	4.00	0.60
Spectral analysis M390	1.98	0.79	0.38	20.37	0.85	4.02	0.53
BOHLER M398	2.70	0.50	0.50	20.00	1.00	7.20	0.70
Spectral analysis M398	2.65	0.55	0.51	20.09	1.00	7.10	0.43

**Table 2 materials-15-07562-t002:** Basic mechanical properties of the investigated M390 and M398 steel [11].

	Modulus of Elasticity [10^3^ N/mm^2^]	Impact KV/Ku [J]	Tensile Strength [MPa]	Proof Strength Rp_0.2_ (MPa)	Thermal Conductivity [W/m°K]	Specific Heat [J/kg°K]	Elongation [%]
M390	227	34	898	172	16.5	480	33
M398	231	35	1078.5	183	15.2	490	31

All data are Q + T state.

**Table 3 materials-15-07562-t003:** Comparison of hardness of HV10 of materials M390 and M398 after tempering.

	Quenching Samples			Quenching Samples with Deep Freezing		
	Tempering Temperature [°C]			Tempering Temperature [°C]		
Material	200 [°C]	400 [°C]	600 [°C]	200 [°C]	400 [°C]	600 [°C]
M390	735 ± 6	717 ± 12	455 ± 34	758 ± 6	833 ± 9	486 ± 28
M398	806.5 ± 8	815 ± 11	506 ± 22	785 ± 9	830 ± 10	480 ± 25

**Table 4 materials-15-07562-t004:** Comparison of Sa parameter of materials, M390 and M398, and ceramic balls, Al_2_O_3_.

Sample	Sa Material [µm]	Sa Ball [µm]	Sample	Sa Material [µm]	Sa Ball [µm]
M390_200DF	0.16 ± 0.05	0.60 ± 0.16	M398_200DF	0.07 ± 0.02	0.45 ± 0.11
M390_400DF	0.10 ± 0.03	0.34 ± 0.09	M398_400DF	0.06 ± 0.02	0.44 ± 0.11
M390_600DF	0.13 ± 0.04	0.41 ± 0.10	M398 _600DF	0.11 ± 0.03	0.44 ± 0.10
M390_200	0.32 ±0.08	0.50 ± 0.13	M398 _200	0.04 ± 0.02	0.49 ± 0.12
M390_400	0.06 ± 0.02	0.76 ± 0.18	M398 _400	0.08 ± 0.02	0.35 ± 0.09
M390_600	0.10 ± 0.03	0.49 ± 0.11	M398 _600	0.02 ± 0.01	0.32 ± 0.09

## Data Availability

Data are available upon request to the corresponding author.

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
