# Peer review of "Analysis of Tribological Properties of Powdered Tool Steels M390 and M398 in Contact with Al2O3"

_materials, 2022, doi:10.3390/ma15217562_

Round 1
Reviewer 1 Report
The paper cannot be accepted for publication mainly due to its extremely low quality. The authors should learn how to properly write a scientific paper.
(1) The presentation is too wordy and lengthy. The authors did not present the paper with a clear logic flow and focus; instead they presented too much information that is not needed, or with limited value, or should be presented in a concise way.
(2) Some results were presented in the experimental while some testing procedures were presented in the results.
(3) It is not sure that information presented in Fig. 6 and Fig. 7 are the results of the authors or from the steel manufacturer.
(4) The important results are extremely limited. Lot of results can be presented with a very short paragraph, instead of so many not-clearly-selected photos/figures.
(5) The paper is lack of a good and deep discussion.
Reviewer 2 Report
General Comment: I want to congratulate the authors for the manuscript titled as “Analysis of tribological properties of powdered tool steels M390 and M398 in contact with Al2O3”. The article is well-written, the results are supported by enough experimental data. The results are worth to be publish. I have some questions/comments:
· Abstract: writing is too generalized. The main theme of this paper is not described in the abstract. Abstract section should be concisely reflected the content and summarize the problem, the method, the results, and the conclusions. Also, please add more qualitative and quantitative results of your work.
· At the last paragraph of the introduction, please clearly show the general outline of the paper and show the importance of the study along with the main aim.
· Each one of the cited references must be discussed individually and demonstrate their significance to your work. Not [3,4,5], should be [3] text what is presented in the manuscript [3] text what is presented in the manuscript [5]. Maybe you should decrease numbers of references.
· All the article is too long. The reader forgets some of the results by the end. Try to shorten. maybe move some results to Supplementary.
· Indeed, there are an impressive amount of results. However, the conclusions section needs to improve with selected and highlighted main findings. In conclusion section, it is necessary to more clearly show the novelty of the article and the advantages of the proposed method. Add qualitative and quantitative results of your work. Please try to emphasize your novelty, put some quantifications, and comment on the limitations. This is a very common way to write conclusions for a learned academic journal. The conclusions should highlight the novelty and advance in understanding presented in the work.
· Language used in the manuscript is generally satisfying. However, writers should pay more attention of singular / plural nouns. Also, they should control the spell check/ punctuation of words and sentences. In addition, spaces should be added between words and numbers. Please check all manuscript for language and misspellings. Such as page 7 line 233 49,05N should be revised as 49.05 N. For example, page 5 line 172 165 -250. Please revisit all manuscript and correct such inconsistencies.
· There is a reference problem. If your work is convenient for this journal’s context then there are many references from this or other journal. The following publications would be helpful:
· The plotted values in all Figures need standard deviations (error bars).
· Some results and discussion sections in the paper lacks theoretical support. Please provide more detailed discussions to support author's discoveries on the research.
-----------------------------------------------------------------------------------------------------------------
The article is interesting but needs to be improved. Authors should carefully study the comments and make improvements to the article step by step. After minor changes can an article be considered for publication in the "Materials".
Reviewer 3 Report
Dear authors, your manuscript on tribological properties of powdered tool steels 2 M390 and M398 brings valuable and interesting results on the use of new steel for processing polymeric materials. The experiments were well designed, and the results and discussions are worth publishing. However, before being publishable, the paper must be revised. I have highlighted the main points I detected below:
-What do you mean by "desintegrate" in line 70?
- Please rewrite for better clarity the sentence in lines 79 - 81.
- Your introduction should end with the novelty statement of your paper. Lines 118 - 137 don't belong to the introduction section. I believe they are part of the Materials and Methods.
- I believe Fig. 3 is also misplaced; it should be your first result.
-I believe Fig. 4 is also misplaced; it should be in the introduction section.
-I believe Fig. 6 and Fig, 7 are also misplaced; they should be in the results section.
Round 2
Reviewer 1 Report
The reviewer still has reservation to accept the manuscript in current form as:
(1) the introduction, especially the 4th paragraph cited papers in a way that is not closed relevant to this study. The authors should digest the information from others' research works and summarize in a logic and concise way.
(2) the paper after revision is still too wordy, too long and presents something not important. For example, "This material was the Al2O3 alu-224 minum oxide ceramic ball, which is used in bearings with a diameter of 6.35 mm. The 225 crystal structure of the α-phase Al2O3, which is a corundum structure, ideally consists of 226 closely spaced planes (A and B planes) of large oxygen anions with a diameter of 0.14 nm 227 arranged in sequence [21]. The cations occupy only two-thirds of the octahedral places of 228 basic distribution. This location consists of three different types of aluminum cationic lay-229 ers, named a, b, and c. Figure 4a shows the complete oxygen stacking sequence. The alu-230 minum layers form a sequence A-a-B-b-A-c-B-a-A-b-B-c-A…One period in this order, i. 231 from c-A to B-c, forms the hexagonal base cell α-Al2O3 [22]. An example of the test, the 232 shape of the ball, and the hardness of both contact materials are shown in Fig. 4b." should be removed from the paper. The paper must be substantially shortened. Figure 2 and Figure 4 should be removed from the paper.
(3) The discussion still lacks of depth. Though the materials are new, the authors should still be able to explain and discuss the results with depth.
Round 3
Reviewer 1 Report
The paper can be accepted now.
Author Response
I thank the reviewer for his professional approach and valuable advice when reviewing the article, which improved the quality of our contribution.